# In-Context Learning Unlocked for Diffusion Models

**Zhendong Wang**[1,2], **Yifan Jiang**[1], **Yadong Lu**[2], **Yelong Shen**[2], **Pengcheng He**[2]
**Weizhu Chen**[2], **Zhangyang Wang**[1], and **Mingyuan Zhou**[1]
[1]The University of Texas at Austin, [2]Microsoft Azure AI

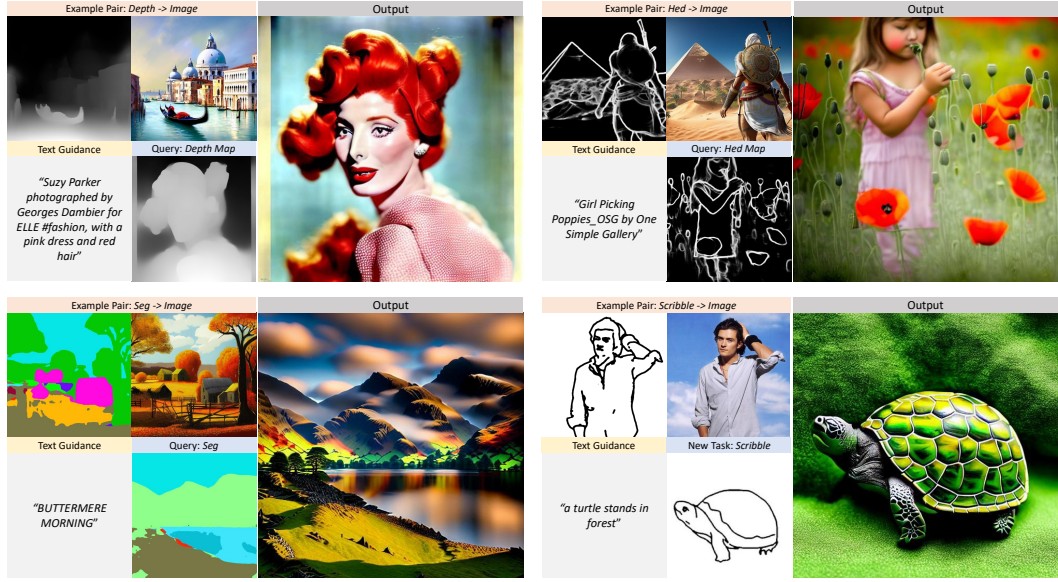

Figure 1: Illustration of the in-context learning ability enabled by our proposed **Prompt Diffusion** for conditioned image generation tasks: With a *prompt* consisting of a *task-specific example pair of images* and *text guidance*, given a new *query image* that aligns in type with the source image in the example pair, Prompt Diffusion can comprehend the desired task and generate the corresponding output image on both seen (trained) and unseen (new) task types.

## Abstract

We present Prompt Diffusion, a framework for enabling in-context learning in diffusion-based generative models. Given a pair of task-specific example images, such as depth from/to image and scribble from/to image, and a text guidance, our model automatically understands the underlying task and performs the same task on a new query image following the text guidance. To achieve this, we propose a vision-language prompt that can model a wide range of vision-language tasks and a diffusion model that takes it as input. The diffusion model is trained jointly on six different tasks using these prompts. The resulting Prompt Diffusion model becomes the first diffusion-based vision-language foundation model capable of in-context learning. It demonstrates high-quality in-context generation for the trained tasks and effectively generalizes to new, unseen vision tasks using their respective prompts. Our model also shows compelling text-guided image editing results. Our framework aims to facilitate research into in-context learning for computer vision. We share our code and pre-trained models at `https://github.com/Zhendong-Wang/Prompt-Diffusion`.

37th Conference on Neural Information Processing Systems (NeurIPS 2023).

# 1 Introduction

Recent advancements in machine learning, particularly in the field of natural language processing (NLP) [54], have led to the development of several state-of-the-art large language models (LLMs), such as BERT [9], GPT-2 [36], BART [21], T5 [38], GPT-3 [5], and GPT-4 [33]. These models have been successfully applied to a wide range of tasks, including sentiment analysis, question answering, machine translation, and text generation, to name a few. An emergent behavior of these LLMs is their ability to learn from context, a phenomenon often referred to as in-context learning. In LLMs such as GPT-3 [5], the in-context learning ability allows them to perform a task just by conditioning on the combination of input-output examples and a new query input, $a.k.a.$ prompts, without optimizing any model parameters. With a proper design of the prompt structure and in-context learning, LLMs can unite the pre-training of multiple language tasks and generalize well to previously unseen tasks.

While in-context learning has been extensively studied in NLP, its applications in the field of computer vision are still limited. To showcase the viability and potential of in-context learning as a standard approach for large-scale vision applications, two significant challenges need to be addressed: 1) Designing an effective vision prompt, which incorporates both domain-specific input-output pairs as examples and image queries as conditions, is more challenging than designing prompts for language tasks. 2) In computer vision, large models are typically trained for specific tasks such as segmentation [22], detection [6], classification [20], self-supervised representation learning [37], class-conditional generation [10], and text-to-image generation [41, 44, 67, 31]. As a result, these large vision models are not designed for in-context learning and lack the flexibility to adapt to new tasks. Several recent attempts [3, 7, 56, 57] tackle these difficulties by following the solutions in NLP. More specifically, a simple visual prompt is built by stitching example images, query images, and output images into one large image, and then a Transformer-based image inpainting model is trained to predict the masked output images [3, 56]. Stitching to large images, however, will dramatically increase the computational cost, especially in high-resolution cases.

Tackling these two challenges, this paper aims to unlock the in-context learning ability of text-guided diffusion-based generative models. We introduce a new model architecture, Prompt Diffusion, to perform in-context learning under a vision-language prompt that could accommodate a wide variety of vision-language tasks. We train Prompt Diffusion jointly on six different vision-language tasks. Specifically, we initially establish a general vision-language task by employing our vision-language prompt:

**prompt:** {*text-guidance, example: (image1 → image2), image-query: image3*} → **target:** *image4*,

where *(image1 → image2)* consists of a pair of vision task examples, $e.g.$, *(depth map → image)*, *text-guidance* provides language instructions for specific tasks, and *image3* is the input image query that aligns in type with *image1* and hence could be a real image or an image condition ($e.g.$, depth or hed map). We subsequently develop Prompt Diffusion, drawing inspiration from the design of Stable Diffusion [69] and ControlNet [41], which can take our vision-language prompt as the input. We finetune Prompt Diffusion from Stable Diffusion [69] checkpoints on six different vision-language tasks, concurrently and uniformly, including three forward tasks: image segmentation/hed/depth map generation, and three inverse tasks: image generation given segmentation/hed/depth maps. We illustrate Prompt Diffusion through examples in Figure 2.

Prompt Diffusion successfully integrates the learning of six different tasks into one vision-language foundation model. Through prompt-based learning, the model can effectively grasp the underlying relationship between the input example pairs. The model can then use this understanding to generate the output image by re-mapping the relationship onto the query image and incorporating the language instructions, as illustrated in Figures 2 and 4. More importantly, the model acquires in-context learning ability by learning across multiple tasks. Remarkably, we observe that Prompt Diffusion not only performs well on the six tasks it has seen during training, but also generalizes effectively across several new unseen tasks, as shown in Figure 5.

Overall, we propose Prompt Diffusion as an initial attempt to unlock the in-context learning ability of text-guided diffusion models. Empirically, we observe that Prompt Diffusion has a promising in-context learning ability on both trained and new, unseen tasks. The success achieved by Prompt Diffusion is to inspire and spur additional research in the field of diffusion-based in-context visual learning. We summarize our main contributions as follows:

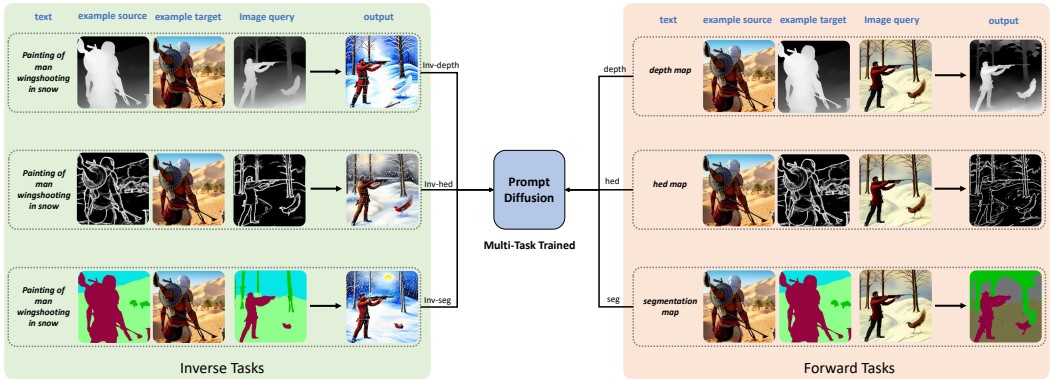

Figure 2: Illustration of Prompt Diffusion trained jointly on six different vision-language tasks. Each gray-dashed box represents a task example: *prompt→output*. The output is a random generation from our trained model given a vision-language prompt, where the query image aligns in type with the source image from the example.

- A novel vision-language prompt design that well supports the integration of various vision-language tasks.
- The Prompt Diffusion model, the first diffusion-based versatile vision-language foundation model capable of in-context learning.
- Demonstration of high-quality in-context generation on both trained tasks and new, unseen tasks.

## 2 Related Work

### 2.1 Diffusion Models

Recent advances in diffusion models have exhibited overwhelming success in generative applications. Diffusion models [47, 49, 17], serving as sophisticated generative models, generate intriguing examples through a step-wise denoising process. These models utilize a forward process that incorporates noise into data distributions and reverses this process to reconstruct the original data. Subsequently, numerous studies [10, 31, 39, 41, 44] are focused on scaling the diffusion models up or improving the training and sampling efficiency [48, 26, 60, 71] to achieve better performance. In particular, LDM [41] is a prominent model that reduces computational costs by applying the diffusion process to a low-resolution latent space, successfully scaling text-to-image generation to web-scale data. Lately, Versatile Diffusion [67] expands the existing single-flow diffusion pipeline into a multi-task multimodal network that handles multiple flows of text-to-image, image-to-text, and variations in one unified model. Many other pieces of research also extend the diffusion framework to various domains, including stable GAN training[59], music generation [18], text-to-3D [34, 2], language generation [24], novel-view synthesis [66], uncertainty quantification [13], reinforcement learning [58], *etc.* Besides generation, recent studies [28, 69, 15, 30, 4, 61, 12, 19] also explore the potential of diffusion models in image and video editing-related applications.

### 2.2 In-Context Learning

Recent advances in large language models (LLMs) have made a substantial impact on multi-modal modeling. Despite this, many of these models [8, 11, 14, 23, 25, 27, 46, 50–52, 55, 53] still necessitate fine-tuning for specific downstream tasks. By contrast, in-context learning, a novel paradigm introduced in generative pretrained transformers (GPT-1/2/3/4 [35, 36, 5, 33]), enables models to rapidly adapt to new tasks using only a handful of prompts. Rather than finetuning on downstream tasks, GPT formulates various language tasks as predicting the future tokens given the context, exhibiting remarkable emergent abilities [62] on unseen tasks without explicit training [29, 42, 65, 70].

Lately, researchers have been working on in-context learning frameworks that encompass computer vision tasks as well. For instance, Alayrac *et.al.* [1] propose Flamingo that combines pretrained vision-only and language-only models and employs paired image-text examples as prompts for few-shot

learning. Bar *et.al.* [3] introduce a universal inpainting model to tackle multiple vision tasks by taking the desired output as a masked region and predicting it based on given examples. Inspired by this approach, Painter [56] further incorporates more tasks to construct a universal task solver, observing emergent abilities on unseen categories if evaluated on in-domain tasks. Subsequently, SegGPT [57] unifies multiple segmentation tasks into a generalist framework and investigates both spatial ensemble and feature ensemble methods for creating prompt examples. Different from previous works, our work is primarily focused on exploring the in-context learning ability of diffusion models.

## 2.3 Prompt Design for Visual Language Models

The idea [5, 36] of transforming various language tasks into a generative problem has motivated a lot of following works to explore how to formulate appropriate generative conditions, which is also known as prompt engineering. Designing different prompts is observed to result in a huge performance gap in many language tasks [32, 63]. Prompt designing also raises much attention in vision areas after the breakthroughs of vision-language models (*e.g.*, CLIP [37]). Instead of directly predicting a one-hot label-assignment vector using a classifier, CLIP can be used as a classifier by using a correct prompt template, such as "A photo of a [$label$]". A following work [72] observes that changing the text template itself matters for the performance of different downstream vision tasks. Other works further propose improving strategies by adding a red circle on the interesting areas of given images [45], or coloring the interesting image patches [68]. While our proposed method also explores vision-language prompt engineering, we further introduce a new interface in the vision backbone that helps take and understand the image prompts.

## 3 Prompt Diffusion and In-Context Learning

In this section, we describe how to perform vision-language prompting on diffusion-based text-to-image generative models. First, we introduce our novel vision-language prompt design in Section 3.1. Then, we show our model design of Prompt Diffusion in Section 3.2. Finally, we demonstrate how we represent different vision-language tasks via our prompting and how we jointly train Prompt Diffusion on multiple tasks in Section 3.3.

### 3.1 Vision-Language Prompt Design

In LLMs, a common way of task-prompting for a specific language understanding task is to provide the trained model with pairs of examples of the target task and the input query [5]. For example, in machine translation, English-to-French, Brown et al. [5] construct the prompt as *{example: sea otter → loutre de mer, query: cheese → }*, and the model will learn from the example in the prompt to translate the query *'cheese'* into *'fromage'*.

Following the convention, we design the vision-language prompt by replacing the text examples with paired image examples and the text query with an image query. We further add one more text input to allow text-guided image generation. By this design, we propose a new input-output pair format that could generalize the input-output configuration of most vision-language tasks. We show it as follows together with one example:

**prompt:** {*text-guidance, example: (image1 → image2), image-query: image3*} → **target:** *image4*,

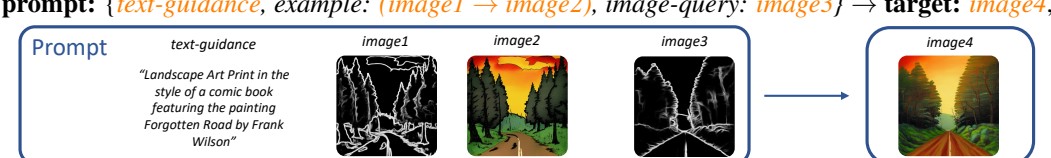

where the *exmaple: (image1 → image2)* informs the model what the target task is and how it could be done through examples, *e.g.*, image generation from the hed map, *text-guidance* guides the model to generate an image conditioning on the given text, and the *image-query: image3*, which aligns in type with *image1*, represents the input to the specific task. The *example* pair could accommodate any image domain transformation tasks, *e.g.*, forward tasks: image → segmentation/depth/hed map, and inverse tasks: segmentation/depth/hed map → image, while the involvement of text-guidance provides additional context to generate the target images. Next, we present Prompt Diffusion, which

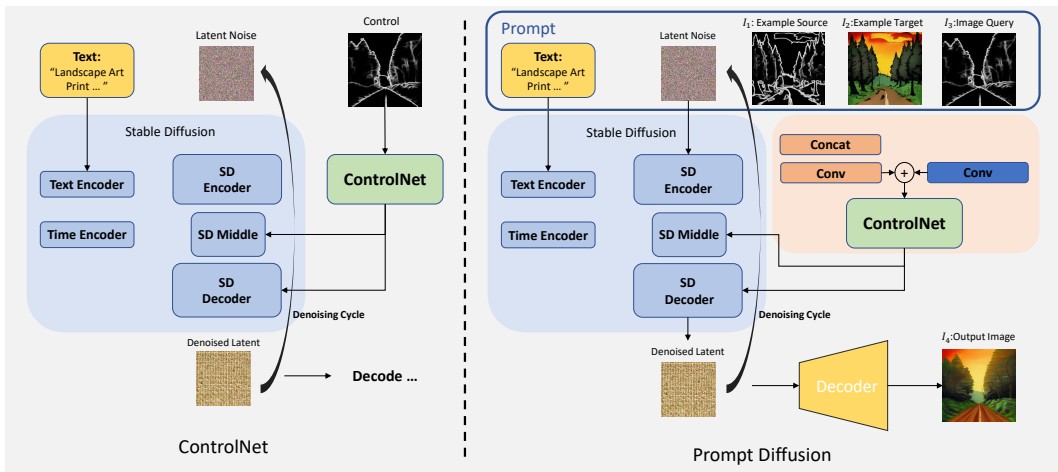

Figure 3: Comparison of the model architectures of ControlNet (Left) and Prompt Diffusion (Right). More details on the network structures of Prompt Diffusion-induced modules can be found in Appendix C.

is a diffusion-based generative model that takes the vision-language prompts as inputs and outputs the target images.

## 3.2 Prompt Diffusion

Motivated by the success of Stable Diffusion [41] and ControlNet [69] on text-to-image and image conditional generation, we build Prompt Diffusion based on the design of ControlNet. Our vision-language prompt is multimodal, incorporating both text and image inputs. For image inputs, we concatenate the example pair of images in the channel dimension, and then project the concatenated example pair and image query into equivalent dimensional embeddings via independent stacked convolutional layers. We compute the sum of the two embeddings and feed it into the ControlNet branch. For the text input, we follow the convention of Stable Diffusion [41] and encode the text input through the pre-trained CLIP [37] text-encoder. The resulting CLIP text embedding is then fed into the Stable Diffusion branch via cross-attention layers. We illustrate Prompt Diffusion in Figure 3.

We keep the Stable Diffusion branch and ControlNet branch the same as their original versions, so that we could finetune Prompt Diffusion from an existing checkpoint without the need to train it from scratch. We finetune Prompt Diffusion from the Stable Diffusion 'v1.5' checkpoint.

## 3.3 In-Context Learning

We consider six different vision tasks in our joint training. We call them forward tasks when the task inputs are clean images and inverse tasks when the task inputs are image conditions, *e.g.*, segmentation/dep/hed maps, as depicted in Figure 2.

**Datasets.** We use the public dataset proposed by Brooks et al. [4] as our base dataset, which consists of around 310k image-caption pairs. Furthermore, we apply the Midas [40] to obtain the depth and normal maps of all images in our dataset. We obtain the image segmentation maps by applying Uniformer [22]. We collect the canny edge maps by the Canny edge detector [6] and hed maps by the HED boundary detector [64]. Note the depth, hed, and segmentation maps are all used for our model training, while the canny edge and normal maps are only used in the testing time for evaluating the in-context learning ability of Prompt Diffusion on new, unseen tasks.

**Inverse Tasks.** We consider three inverse tasks: depth maps to images, hed maps to images, and segmentation maps to images. For any image-caption pair in the dataset, $(I_1, C_1)$, we first sample a random task and another random image $I_2$ to create the example pair, *e.g.*, $(\mathrm{HED}(I_2), I_2)$. Then, we build the vision-language prompt with the example pair, the caption, and the image condition that is consistent with the task specified by the example pair. The image $I_1$ is the denoising target of Prompt Diffusion. One complete example of the inverse-hed task is shown as follows:

**prompt:** *{text-guidance: $C_1$, example pair: [HED($I_2$) → $I_2$], image-query: HED($I_1$)}* → **target:** $I_1$ .

By replacing the hed maps with the other two image conditions, we could obtain the vision-language prompts for all the three inverse tasks.

**Forward Tasks.** We also consider three forward tasks (image processing tasks): images to depth maps, images to hed maps, and images to segmentation maps. We follow the same rule of inverse tasks to create the vision-language prompts. Note here the tasks are flipped, so we flip the order inside both the example pair and the query-target pair. The captions of images are not necessary for image processing tasks, so we use a task-specific fixed text label for each task, such as 'hed maps'. We show one complete example of the forward-hed task as follows:

**prompt:** *{text-guidance: 'hed maps', example pair: [$I_2$ → HED($I_2$)], image-query: $I_1$}* → **target:** HED($I_1$) .

**Joint Training.** We train Prompt Diffusion jointly on these six different vision-language tasks. For simplicity, we train our model on these six tasks uniformly at random. Specifically, each minibatch data contains randomly sampled vision-language prompts and the corresponding target images from randomly selected tasks. Joint training over these different tasks unlocks the in-context learning ability of diffusion models, as shown in Figures 4 and 5. In order to apply classifier-free guidance (CFG) [16], we randomly drop $10\%$ text guidance during training.

## 4 Experiments

In this section, we conduct extensive experiments to illustrate the power of Prompt Diffusion as a strong versatile vision-language foundation model that is capable of in-context learning. We first show that Prompt Diffusion performs well with multi-task training in Section 4.1. Then, we show that Prompt Diffusion has promising in-context learning ability and could generalize well to new, unseen tasks in Section 4.2. We show that Prompt Diffusion supports controllable and text-guided image editing in Section 4.3. Finally, we conduct an ablation study on the three main components of our vision-language prompt in Appendix A.

**Implementations.** We implement Prompt Diffusion upon the codebase of ControlNet [69]. As illustrated in Figure 3, we implement both the visual-example-pair encoder and image-query encoder via stacked convolution layers. See the details of these layers in Appendix C. We load the weights from Stable Diffusion checkpoint 'v1.5' for finetuning. We fix the learning rate at $1 \times 10^{-4}$ and accumulate gradients every 4 mini-batches with batch size 512. The datasets we used and how we augment the original datasets are described in detail in Section 3.3. We train our model on $8 \times$A100 Nvidia GPUs for 5000-20000 steps. Longer training helps the model perform better on the finetuned dataset while may perform worse on out-of-distribution images. The results shown in the paper are based on the 5000-step finetuned checkpoint.

**Evaluation.** We conduct both qualitative and quantitative evaluations to assess the performance of Prompt Diffusion. We provide comprehensive qualitative evaluations to measure Prompt Diffusion's in-context learning capabilities. To assess generation quality, we provide quantitative results measured using the zero-shot Fr'echet Inception Distance (FID) for inverse tasks and Root Mean Square Error (RMSE) for forward tasks, as detailed in Section 4.4. The test dataset is composed of random images from the ControlNet [69] example images, DreamBooth [43] data benchmark, and the test split of our base dataset [4]. During the inference stage, one additional random image from our dataset is used to construct the example pair for each specific task.

### 4.1 Multi-Task Learning

We jointly train Prompt Diffusion on three inverse tasks, including inverting depth/hed/segmentation, and three forward tasks, including extracting depth/hed/segmentation maps. Note ControlNet [69] aims to enhance pretrained image diffusion models with task-specific conditions (*e.g.*, depth maps). We finetune a ControlNet, from the same Stable Diffusion checkpoint, on our dataset for each inverse task independently as our baseline, and we refer to it as CN(FT). We qualitatively evaluate Prompt Diffusion on the six trained tasks in Figure 4.

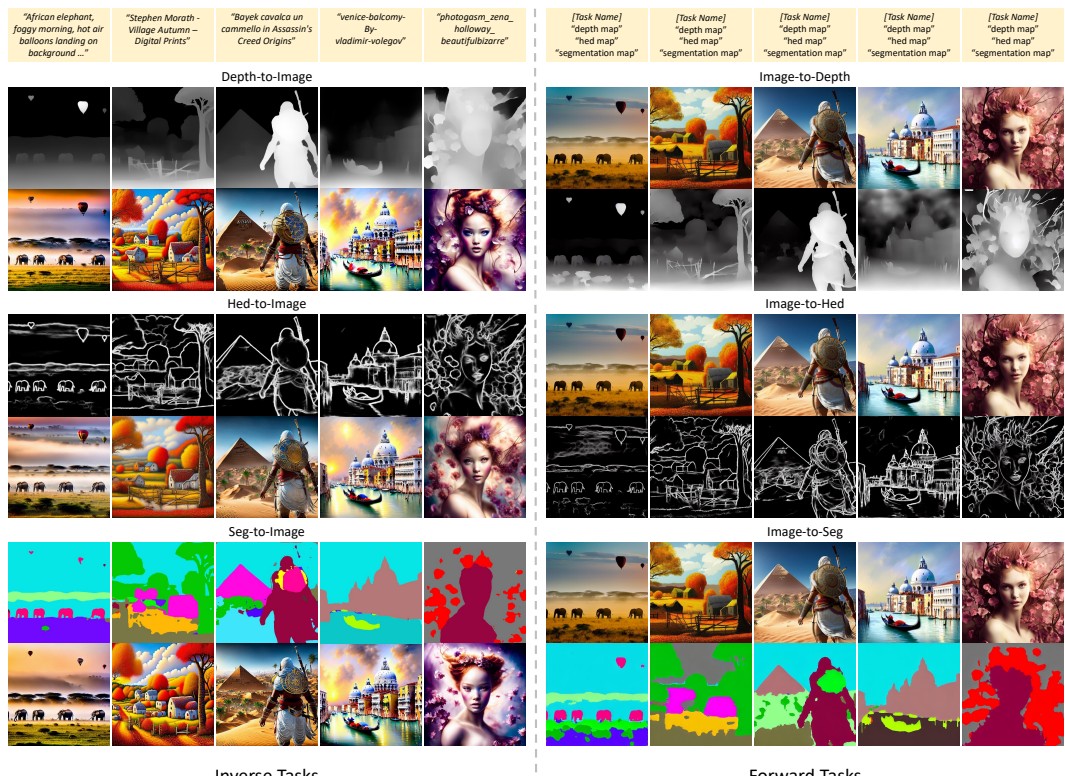

Figure 4: **Qualitative Results of forward tasks and inverse tasks.** We show examples of applying our Prompt Diffusion on the six trained tasks (inv-depth/hed/seg and forward-depth/hed/seg).

We observe that Prompt Diffusion could not only generate high-fidelity images based on different task-specific conditions for the inverse tasks, but also generate detailed and rational image conditions if it is given the real images for the forward tasks. For an inverse task, when the image conditions already sketch the main components of the output images, such as depth and hed maps, Prompt Diffusion could successfully maintain the shape consistency in the output images and generate diverse images with different random seeds and text guidance. When the image conditions are coarse like segmentation maps, Prompt Diffusion could generate more imaginary images (*e.g.*, the fairy with a personalized hair style in the third block). For forward tasks, we empirically find Prompt Diffusion generally produces even more detailed depth and hed maps compared to these maps in the original dataset. The produced segmentation maps are comparable to the ones used as the targets during finetuning.

Note ControlNet [69] is designed for image generation with image conditions. Hence, we compare Prompt Diffusion with CN(FT) on the three inverse tasks in Figure 10 of the Appendix. The results show that Prompt Diffusion performs comparably well to CN(FT) which is trained specifically for each individual task. We further evaluate the generation ability of CN(FT) in Figure 11 of the Appendix. We directly infer CN(FT) on new tasks and observe a large number of failures. This validates the necessity of multi-task training for Prompt Diffusion to achieve the capability to generalize across a wide variety of tasks.

## 4.2  In-Context Learning Ability

We evaluate Prompt Diffusion on tasks that are outside of our supervised training dataset to further study its generalization ability. We first select three unseen tasks, inv-scribble (image generation from personalized scribbles), inv-edge (image generation from canny-edge maps), and inv-normal (image generation from normal maps). The model is given a pair of image examples that illustrates how to perform a desired task, and is asked to re-conduct the task onto new image query inputs with text guidance.

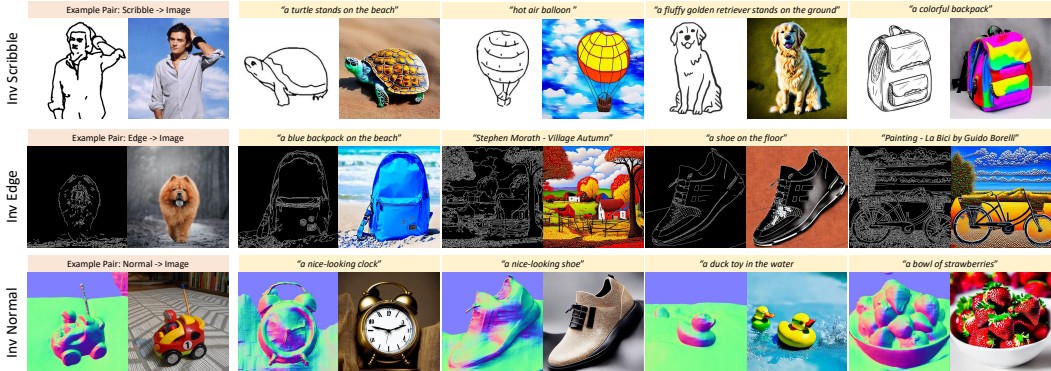

Figure 5: **Generalization to New Tasks.** We show Prompt Diffusion has a promising generalization ability to new, unseen tasks, such as Inv-Scribble, Inv-Edge (CannyEdge), and Inv-Normal (Normal map).

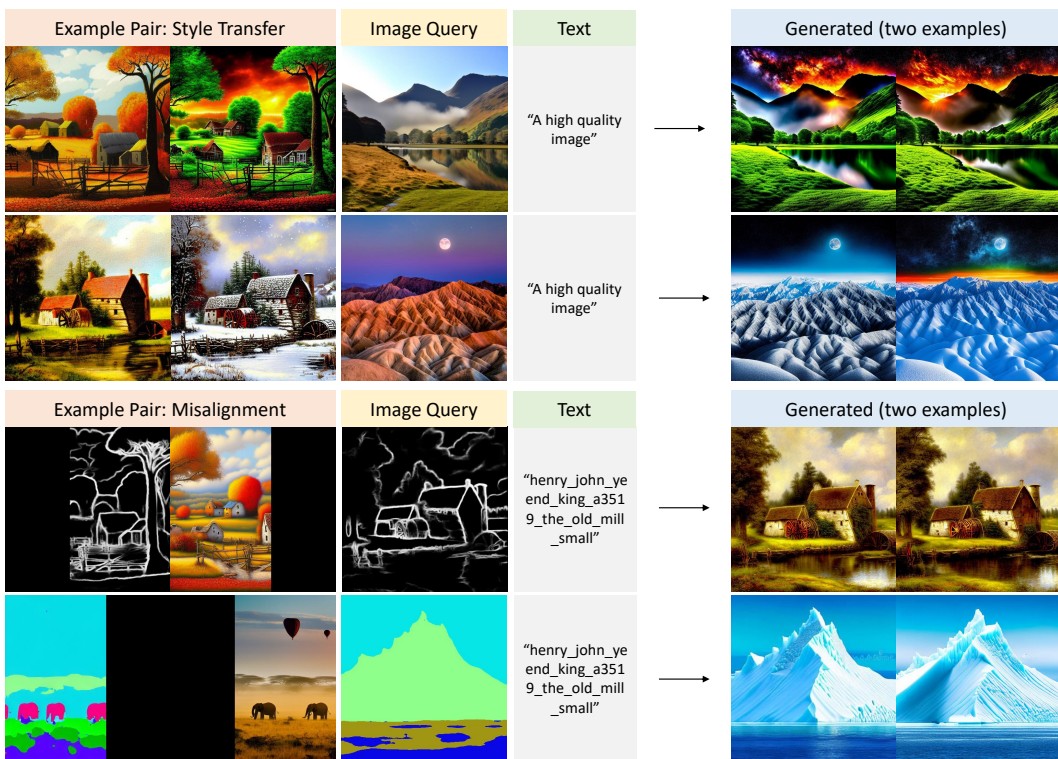

Figure 6: We perform Prompt Diffusion on two completely new tasks: style transfer and misaligned input-output example pairs. The first two rows show the style transfer results while the last two rows show the misaligment results.

We show the qualitative results in Figure 5. Prompt Diffusion successfully learns the underlying correlation between the example pairs and re-maps the relationship onto query and output pairs, without the need of training or finetuning on the three new domains. The image generation is also well controlled by text guidance as shown in the results.

In Figure 6, we further evaluate Prompt Diffusion on two distinct tasks: style transfer and misaligned input-output example pairs. For the first task, as shown in the initial two rows, Prompt Diffusion is given a pair of image style transfer examples. It is then tasked with applying this style transfer to a new target image, accompanied by a general text description, such as "A high quality image." The findings indicate that Prompt Diffusion is not only capable of discerning the underlying task from the visual example pairs but can also adeptly carry it out on fresh queries. For the second task, detailed in

the final two rows, we investigate the impact of pixel alignment in the example pairs. Even when various regions in the example pair are concealed, resulting in pixel misalignment, Prompt Diffusion manages to execute the intended task proficiently without any significant disruption.

## 4.3 Image Editing

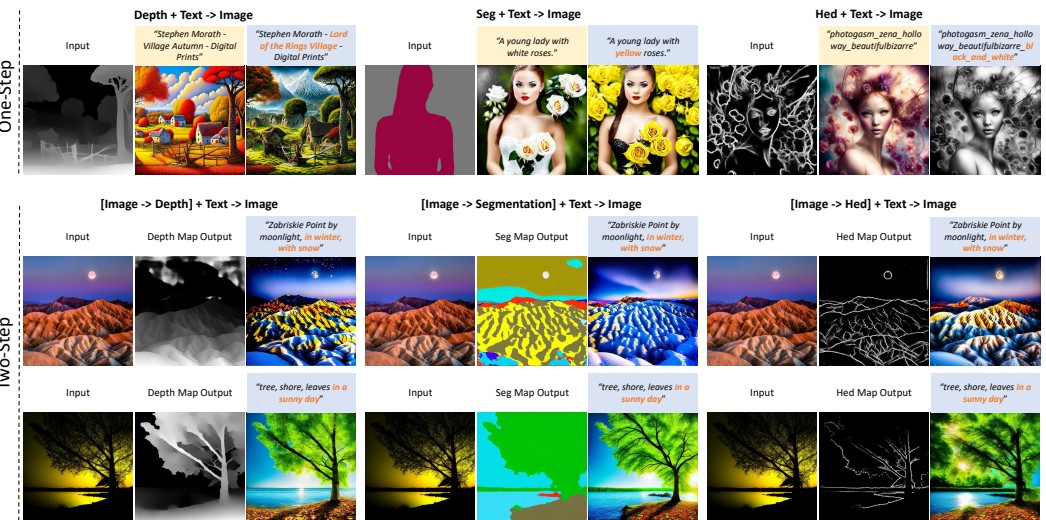

Figure 7: **Image Editing Results.** We demonstrate that Prompt Diffusion excels in image editing tasks through two strategies: one-step and two-step. One-step: when only the image condition, *e.g.*, depth/seg/hed maps, is given, our model would edit the condition for generation according to the text descriptions. Two-Step: when the original image is given, we first do forward sampling to sample image conditions and then conduct inverse tasks to edit the condition with text guidance. Two-Step provides more controllable editing.

We show that Prompt Diffusion has strong image editing ability in two ways. First, if only the conditions of images are given, *e.g.*, depth/segmentation/hed maps, Prompt Diffusion is able to synthesize images based on the input image conditions and additional text descriptions. Second, if one base image is given, we could apply a two-step strategy to achieve controllable image editing: We first generate the image conditions and then generate edited images based on both the generated conditions and text descriptions, all within one model: Prompt Diffusion. As shown in Figure 7, with the help of image conditions, Prompt Diffusion is capable of performing consistent modifications specified by the text guidance input without worrying about undesired excessive changes.

## 4.4 Quantitative Results

We compare Prompt Diffusion with CN(FT) quantitatively. For inverse tasks, we measure the zero-shot FID on the test split of our base dataset [4]. We use the corresponding image condition, text input, and example pair as the input for either Prompt Diffusion or CN(FT), and generate 10k random images where each input produces one image. We then measure the FID using the 10k generated samples with the corresponding 10k test images used as reference. For forward tasks, we measure the zero-shot RMSE between the generated image conditions (e.g., depth/hed/segmentation maps) and the corresponding ones in the test set, where all image conditions are rescaled to $[0, 1]$ for computing the RMSE. Note for both Prompt Diffusion and CN(FT), we finetune them with the same number of 5000 steps. We show the quantitative comparison in Table 1. We observe that Prompt Diffusion has a comparable or even better performance compared to CN(FT).

## 5 Conclusion and Discussion

In this study, we introduce Prompt Diffusion, a novel in-context visual foundation model that leverages diffusion-based techniques to generate images from vision-language prompts. Our model is designed to be flexible and compatible with various vision-language tasks, allowing it to accommodate multiple use cases. We train Prompt Diffusion on six different vision-language tasks and conduct

Table 1: Zero-Shot Quantitative Results. We provide the FID comparison for inverse tasks and RMSE comparison for forward tasks on our test dataset.

| Methods | FID ↓ (Inverse Tasks) | | | RMSE ↓ (Forward Tasks) | | |
|---|---|---|---|---|---|---|
| | Depth-to-Image | Hed-to-Image | Seg-to-Image | Image-to-Depth | Image-to-Hed | Image-to-Seg |
| CN(FT) [69] | 19.81 | **13.07** | 20.71 | **0.20** | 0.18 | 0.36 |
| Prompt Diffusion (ours) | **18.60** | 13.35 | **19.46** | 0.21 | **0.14** | **0.31** |

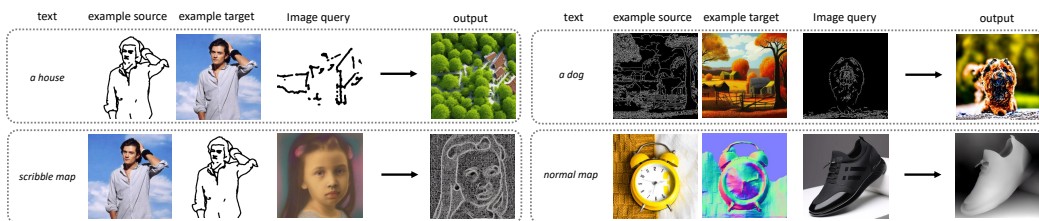

Figure 8: We present some failure cases of Prompt Diffusion when generalizing to new, unseen tasks under ambiguous text guidance.

extensive experiments to evaluate its performance. Our results demonstrate that Prompt Diffusion has a promising generalization ability, not only on trained tasks but also on unseen ones. Additionally, Prompt Diffusion enables controllable image editing, including style, artistic medium, and contextual changes, without introducing excessive undesired modifications.

While our model has demonstrated notable successes in in-context learning, it is important to acknowledge its limitations. Currently, the visual scope of our model is restricted by the limited number of real-life images in the training dataset used for finetuning. As our base dataset [4] is synthesized by Stable Diffusion with proper filtering, it is difficult for the model to generate high-fidelity, real-life images. Moreover, our model is only trained jointly across six pre-defined tasks, and we believe that expanding the range of joint tasks would enhance Prompt Diffusion's in-context learning capability. As shown in the last row of Figure 9, it is difficult to predict the behavior of our model when the query image is misaligned in type with the source image in the example pair, and as shown in Figure 8, there are instances where our model struggles with new tasks. Prompt Diffusion is also limited when it comes to executing physical transformation tasks like rotation and flipping without additional fine-tuning specific to these tasks. To unlock more of Prompt Diffusion's in-context learning ability, it is possible to change the finetuning setting to training from scratch, but this would require significant computational resources, which presently limits its feasibility for us.

Prompt Diffusion represents a pioneering effort in unlocking the in-context learning capacity of diffusion-based models. We hope that this work will encourage more researchers to explore the potential of diffusion-based in-context learning and contribute to the advancement of this exciting field.

## Acknowledgments

Z. Wang and M. Zhou acknowledge the support of NSF-IIS 2212418, NIH-R37 CA271186, and the NSF AI Institute for Foundations of Machine Learning (IFML).

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

# Appendix

## A   Ablation Study

We conduct an ablation study to investigate the effects of vision-language prompt, which consists of the example pair, text guidance, and image query. We show the generation results in Figure 9. In the first row, we keep the text guidance and image query fixed, while we use different random images to construct the example pairs. The results suggest that Prompt Diffusion is not sensitive to what images are used to construct the example pairs, while it cares about the underlying correlation between the images inside the example pairs. In the second row, compared to the first row, we only modify the image query part, and we could see the output images are dramatically changed but consistent with the new image queries. In the third row, apart from the first row, we modify the text guidance and observe the output images are modified accordingly. In the last row, we investigate the impact of using an image query that is intentionally set to be misaligned in type with the source image of the example image pair. Our results show that in such cases, the model's behavior often becomes difficult to predict. Overall, with our vision-language prompt where the image query is aligned with the source image in the example image pair, the example pair tells the model what the specific task is, the image query provides the image domain condition for the generation, and the text guidance allows diverse semantic modifications.

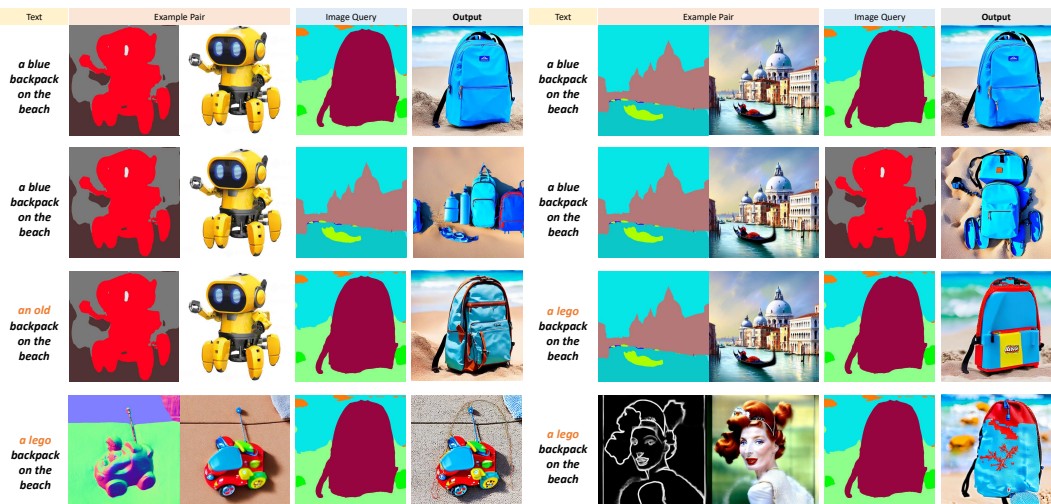

Figure 9: **Ablation study** for different variations of the vision-language prompt in Prompt Diffusion. The last row shows when intentionally mismatching the type of the query image and that of the source image in the example pair, the behavior of Prompt Diffusion becomes difficult to predict.

## B   Comparison to ControlNet

We qualitatively compare Prompt Diffusion with ControlNet [69]. We follow the guidance of ControlNet, finetune a ControlNet, from the same Stable Diffusion checkpoint, on our dataset for each inverse task independently as our baseline, and we call it CN(FT). We show the comparison results in Figure 10. We observe the jointly trained Prompt Diffusion performs comparably well to the independently trained ControlNet, which implies no significant performance change with multi-task learning applied.

We further evaluate the generation ability of the task-specific ControlNet in Figure 11. We directly apply the independently finetuned ControlNet on one new task during inference. We could observe a large number of failures when directly applying ControlNet to new, unseen tasks.

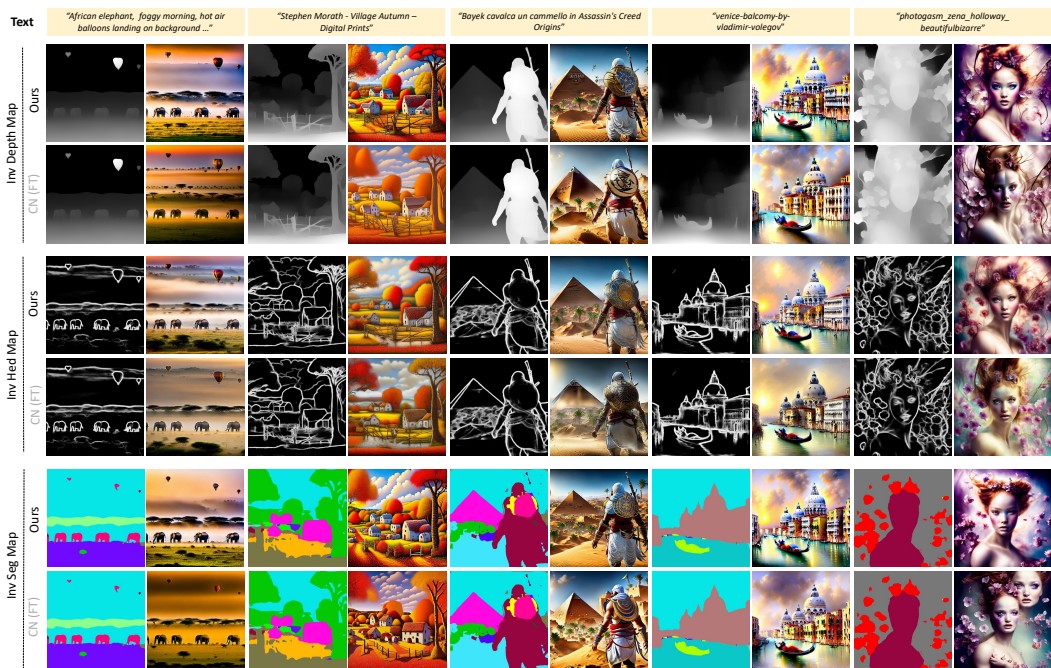

Figure 10: We show the comparision of Prompt Diffusion with CN(FT), ControlNet that is finetuned specifically for each individual task, on the three inverse tasks.

## C    Model Architecture

We build Prompt Diffusion based on ControlNet [69]. We show the detailed model architecture in Figure 12. Note we lock the parameters of Stable Diffusion encoder blocks for finetuning to inherit the encoder capability that Stable Diffusion learns from large-scale pre-training. All the other parameters are open for finetuning. The Stable Diffusion branch takes images in the latent space as inputs, while the ControlNet branch takes the original images as inputs. The pretrained image encoder and decoder, which map images to latent space and re-map latent images to original images, are not shown in the illustration. The latent images are in four channels and eight times smaller in both height and width.

As mentioned in the main paper, our ControlNet branch takes three images as inputs. Two of them form the example pair, which is first concatenated in the RGB channel and then encoded by stacked convolution layers. The third image represents the image query and is encoded with independent stacked convolution layers. We show the architecture of the stacked convolution layers in Figure 13. We encode the example pair and the image query to the same dimensional latent embeddings and then sum them up as the inputs for the ControlNet branch.

## D    Potential Social Implications

The development of image generation models has the potential to have negative social impacts. These models can create realistic images that are difficult to distinguish from real photographs, which could lead to a proliferation of fake images used for malicious purposes. For example, these models could be used to create fake images of people committing crimes or engaging in immoral behavior, which could damage reputations and lead to false accusations. Additionally, the widespread use of image generation models could contribute to the erosion of trust in visual evidence, which could have implications for legal proceedings, journalism, and other fields that rely on visual evidence. There is also a risk that these models could perpetuate biases and stereotypes, as they may learn from biased datasets and generate images that reinforce those biases. It is therefore important to carefully consider the ethical implications of image generation models and take steps to mitigate their potential negative impacts.

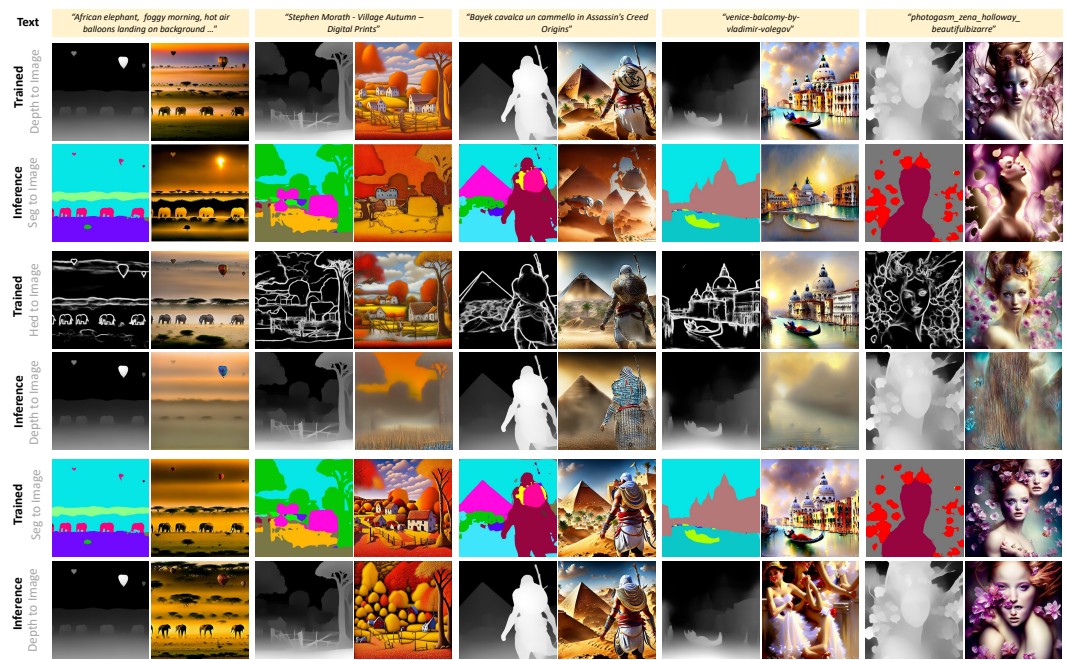

Figure 11: We show that a ControlNet finetuned for an individual task on the dataset fails to generate high-quality images for another task, which validates the necessity of multi-task training.

# E   More Examples

We show more exmaples of applying Prompt Diffusion here.

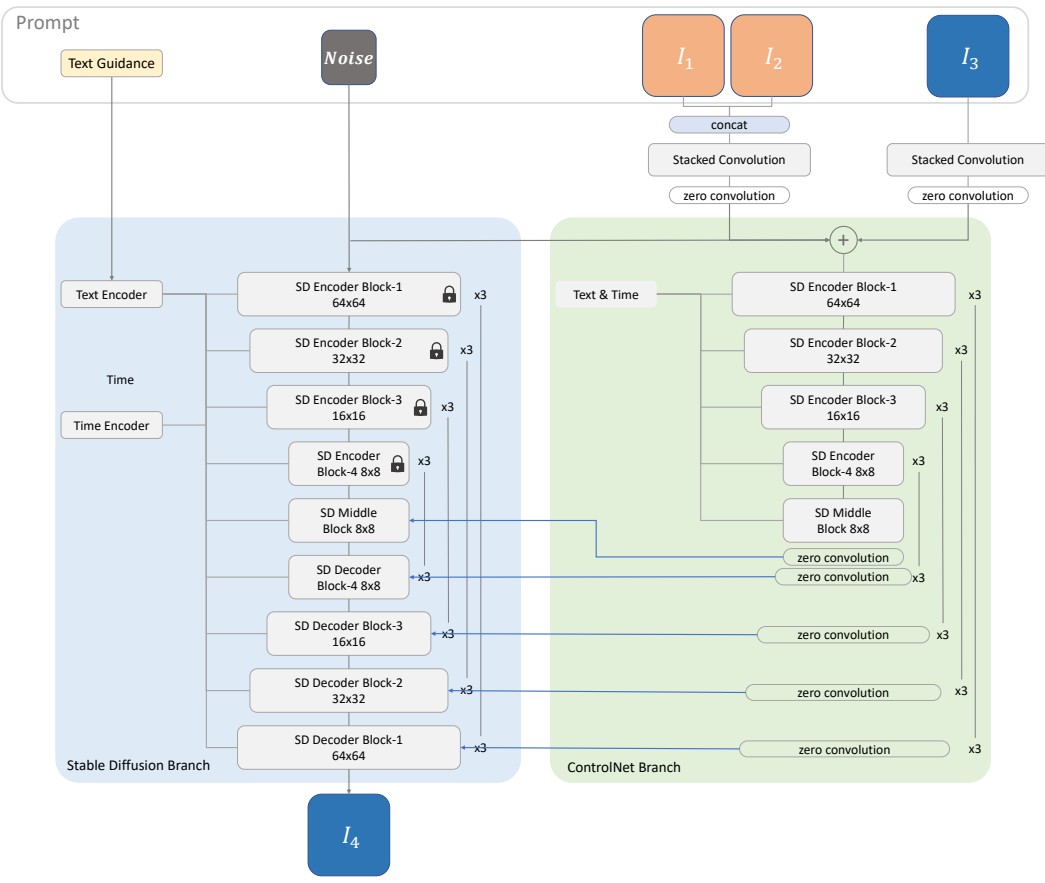

Figure 12: We show the detailed model architecture of Prompt Diffusion.

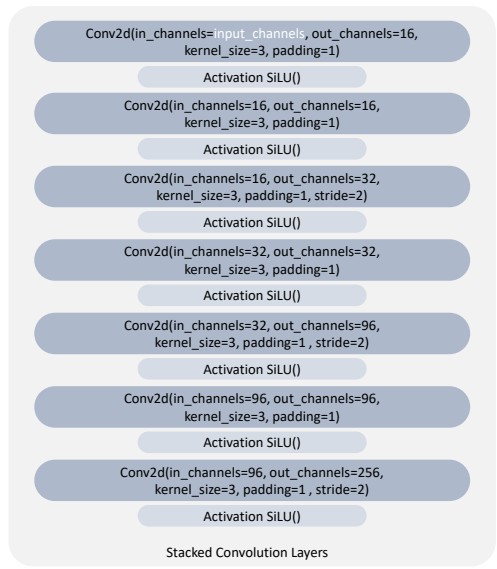

Figure 13: Stacked Convolution Layers.

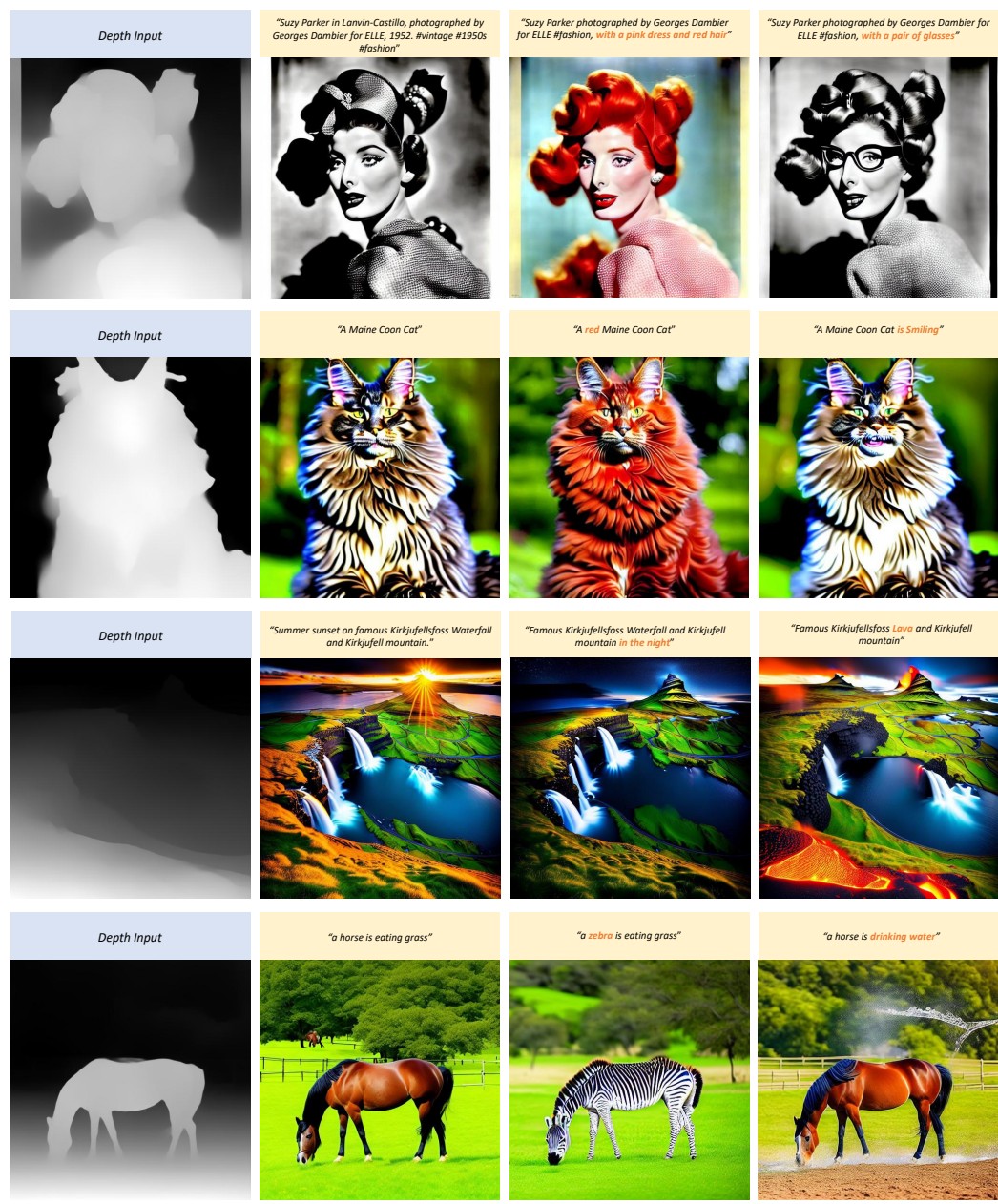

Figure 14: More Depth-to-Image Examples of Prompt Diffusion.

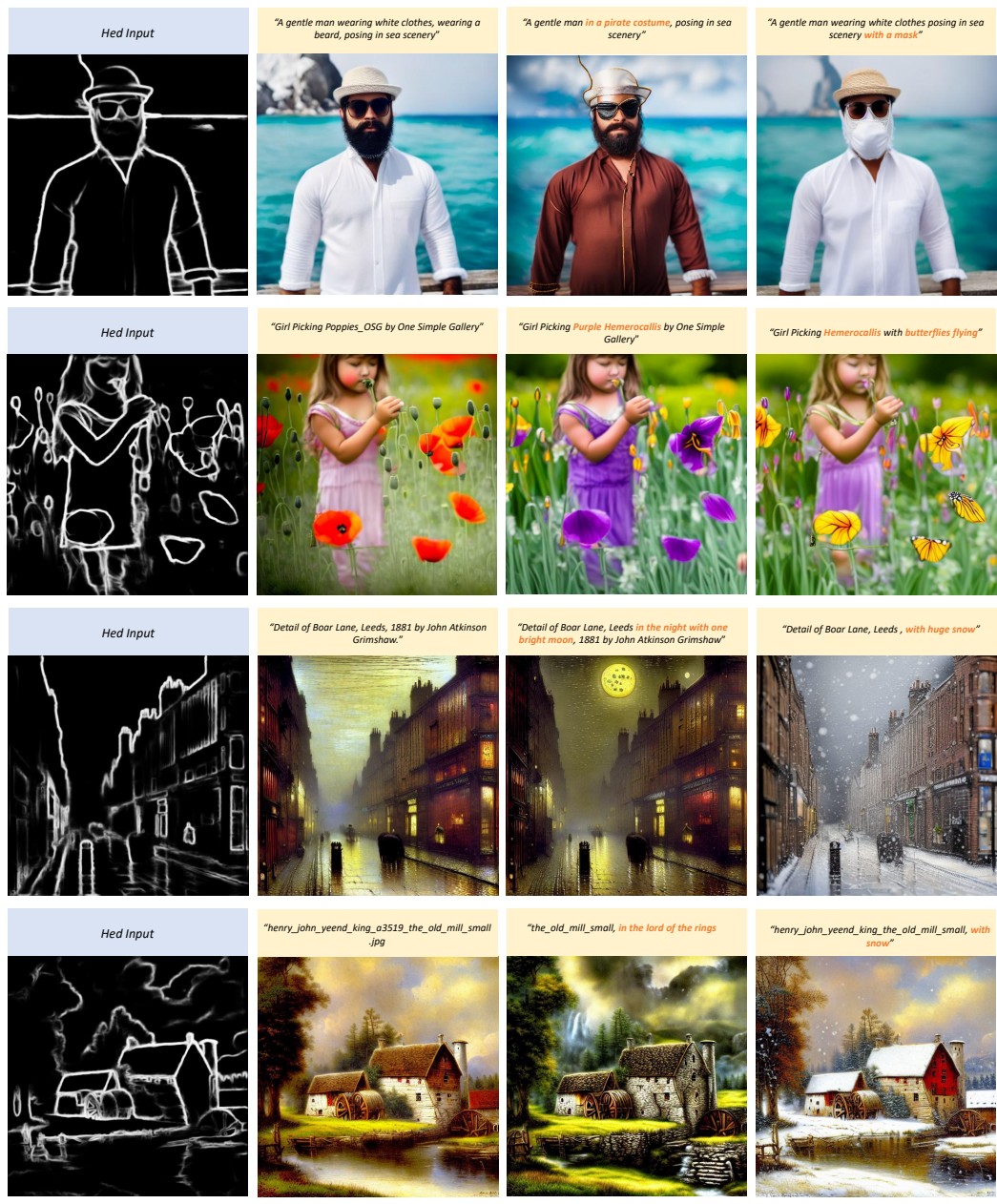

Figure 15: More Hed-to-Image Examples of Prompt Diffusion.

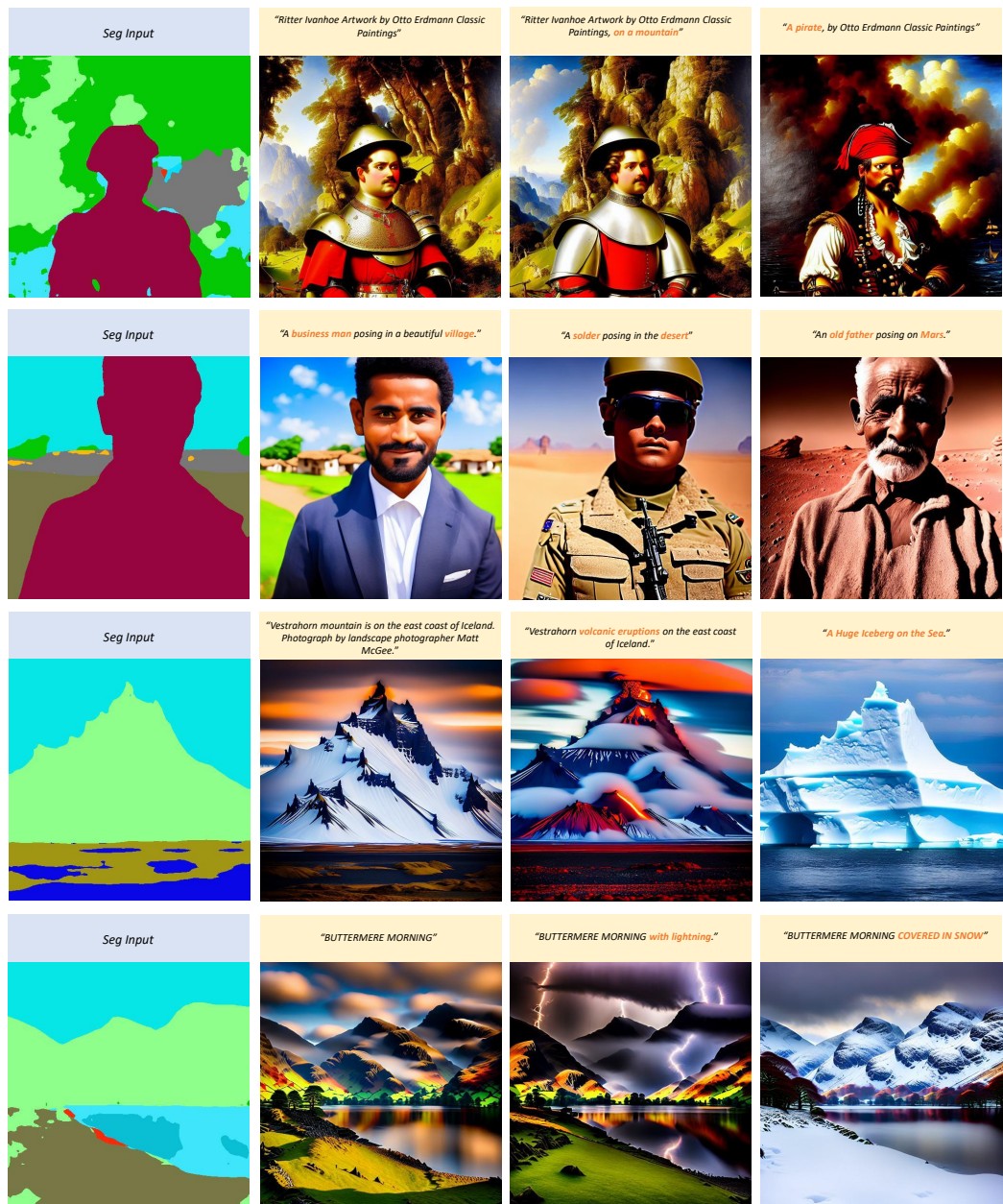

Figure 16: More Seg-to-Image Examples of Prompt Diffusion.

