# OpenReview forum: "In-Context Learning Unlocked for Diffusion Models"
_NeurIPS.cc/2023/Conference — NeurIPS 2023 spotlight_

### Official Review · Reviewer_Gb8x · 2023-07-04

**Soundness:** 2 fair
**Presentation:** 3 good
**Contribution:** 3 good
**Rating:** 5
**Confidence:** 4

**Summary:**

The authors propose an image generation framework. Given the textual description of a task and an input-output example pair, Prompt Diffusion can inpaint the missing output in a consistent way like in the input-output example and the textual guidance instruction. Prompt Diffusion is trained (in a supervised manner) over 6 different tasks and shows some generalization to new tasks and demonstrates emerging image editing capabilities.

**Strengths:**

- Overall, I think the problem setting of visual prompting conditioned on both visual examples and text is novel and interesting.
- The paper is well written and the presentation is clear
- By starting from pretrained diffusion models, the authors bring the advantages of visual quality and text guidance to the prompt diffusion framework.

**Weaknesses:**

My main concern is with the framing of the paper, claiming in-context learning and generalization to new tasks. I think this is a very broad claim which is not backed up by the results. Please see the questions box below for more elaborate feedback/questions.

1. The authors claim the model is "capable of in-context learning" and "generalization to new tasks". Given the empirical results here I find this is hardly convincing.
2. The model is trained on a discrete number of tasks, so it is unclear whether visual examples are actually needed or if text-only guidance is enough.
3. In general, I think more results are necessary to establish the motivation for the new proposed framework.

---

Post rebuttal: given that the authors decided to acknowledge the limitation we discussed, I've decided to update my score.

**Questions:**

1. The model is "capable of in-context learning" and "generalization to new tasks".
* Can the model extend to different input-output structures in the test time? For example, by utilizing more than one visual example?
* Given that the input-output examples are concatenated in channels dimension, can the model extend to input-outputs that are *not* aligned pixel-wise?
* Considering edge prediction as in-context learning after supervised training for depth/hed is a stretch. These tasks are super similar. Similarly, with normals given depth/segmentation supervision.
* Given the previous points, is it possible that the model performs in-context learning of style (Pix2Pix prompting)? I find this claim more accurate, and I think this is still novel.

2. The model is trained on a discrete number of tasks
* Are visual examples actually needed? How does the model perform with standard image input image output with text guidance? (similar to InstructPix2Pix).

3. In general, I think more results are necessary to establish the motivation for the new proposed framework. For example:
* Which modality instructions are more important better? Is it text? image?
* What are the computation tradeoffs? I assume the text is more efficient than images and easier to obtain. Textual prompts are faster, easier to obtain, and have less memory overhead.

**Limitations:**

I think the authors should consider emphasizing the following limitations which were not mentioned:
* the current framework is not very flexible and it assumes a fixed structure of the input-output and query image.
* is the in-context learning limited to style interpolation? (e.g normals based on depth/segmentation, edges instead of depth/hed)

---

> ### Author Rebuttal · Authors · 2023-08-07
>
> We thank Reviewer Gb8x for acknowledging the novelty of our proposed framework, appreciating our presentation, and proposing constructive feedback. Below, we address the concerns raised in your review point by point.
>
> > Claiming in-context learning and generalization to new tasks is very broad.
>
>   We acknowledge that in-context learning within the realm of vision is still a step behind the impressive advancement in large language models. These language models benefit from extensive datasets and the capacity to tackle a wide range of tasks with remarkable flexibility. Considering that the datasets and designed tasks are still in the exploratory process, we believe it will be good for the community to not set the bar unrealistically high by comparing the accomplishments of in-context learning progress in language models against those in vision models.
>
>   The pursuit of in-context visual learning with the ability of generalizing to new tasks has attracted increasing popularity within the vision research community. Our innovation, Prompt Diffusion, distinguishes itself from prior works [1,2] by incorporating this pivotal attribute into the state-of-the-art generative framework, Diffusion Models. This amalgamation enables a robust capability to effectively process diverse input samples and manifests a powerful capacity for comprehending distinct modalities encompassing both images and text. Consequently, in contrast to solely relying on image-visual prompts, Prompt Diffusion accommodates multi-modal vision-language prompts, culminating in the generation of high-resolution and photo-realistic images. Quoting Reviewer foSi, the external dimension brings more versatility and applicability to our model.
>
>   Prompt-Diffusion starts from a diffusion-based approach to solve the in-context visual learning problem. We are optimistic that through meticulous architecture design and the utilization of a diverse and comprehensive dataset, encompassing a wide range of tasks, it could significantly enhance the performance of Prompt-Diffusion in the context of visual learning.
>
>   Following the reviewer's suggestion, we plan to add a limitation discussion in our revised manuscript, as shown in the following. “It is essential to acknowledge that the field of in-context learning in vision is currently in a developmental phase and encounters specific challenges in aligning with the accomplishments attained by large language models. Despite the progress achieved through Prompt-Diffusion, it is conceivable that constraints may exist concerning the extent and adaptability of tasks that can be proficiently accommodated. Going forward, further exploration and improvements in architecture and dataset diversity are needed to overcome these limitations and advance the progress of in-context visual learning.”
>
> Question 1
> - > Q1.a Can the model extend to different input-output structures in the test time? For example, by utilizing more than one visual example?
>
>   Our current design has a fixed number of stacked convolutional layers to encode example and query images, so the use of CNN based encoder limits the number of visual examples that can be plugged into. However, modifying the encoder to a ViT based model would potentially bring more flexibility in test time for encoding visual examples.
>
> - > Q1.b Can the model extend to input-outputs that are not aligned pixel-wise?
>
>
>   For the current designed tasks, the input-outputs are aligned pixel-wise, which is determined by the dataset that we use. If we have a dataset that supports unaligned input-output pairs, then training on the data would make it work.
>
>
> - > Q1.c The conducted tasks are super similar.
>
>
>   Note the tasks look similar from human knowledge but that doesn’t mean the model trained on one task could easily transfer to the others. We conducted one ablation study in Figure 10. We show that a ControlNet finetuned for an individual task on the dataset easily fails to generate high-quality images for another task.
>
> - > Q1.d Is it possible that the model performs in-context learning of style (Pix2Pix prompting)?
>
>   We follow the convention in the literature in claiming in-context image generation.
>
> Question 2
>
> - > Q2.a The discrete number of tasks is limited by the dataset that we have.
>
>   We believe utilizing a larger and more diverse in-context learning dataset could significantly strengthen Prompt-Diffusion.
>
>
> - > Q2.b Are visual examples actually needed?
>
>   Yes, see our ablation study in Figure 8, where we investigate the effects of our vision-language prompt, which consists of the example pair, text guidance, and image query.
>
> - > Q2.c How does the model perform with standard image input image output with text guidance? (similar to InstructPix2Pix).
>
>
>   Our model can take standard images as inputs for forward tasks, such as depth/hed/segmentation map generation. For image editing (similar to Instruct Pix2Pix), we show examples in Figure 6, and the two-step strategy uses standard images as inputs.
>
> Question 3
>
> - > Q3.a Which modality instructions are more important better? Is it text? Image?
>
>   We provide ablation study in Figure 8, which validates that both texts and images are important during image generation.
>
> - > Q3.b What are the computation tradeoffs?
>
>   Yes, text is more efficient than images and easier to obtain.
>
> Limitations
>
> - > The current framework is not very flexible
>
>   The current structure of the input-output and query image is fixed, but we think it could be extended to a more flexible version and we left that for future work.
>
> - > Is the in-context learning limited to style interpolation?
>
>   We follow the convention in the literature in claiming in-context image generation.
>
> References:
>
> [1] Bar, Amir, et al. "Visual prompting via image inpainting." NeurIPS 2022.
>
> [2] Wang, Xinlong, et al. "Images speak in images: A generalist painter for in-context visual learning." CVPR 2023.

---

> > ### Comment · Reviewer_Gb8x · 2023-08-12
> > **Response to authors**
> >
> > I'm very disappointed with the author's rebuttal which has not addressed my main critic. I still believe the claim for in-context learning is unwarranted and not backed up by empirical evidence. In-context learning is an emergent capability of large neural networks and I don't see anything emergent here.
> >
> > Specifically:
> >
> > 1. In-context learning was tested on tasks there are very similar to the ones they were trained on (in a completely supervised manner). Claiming that in-context learning emerges on "unseen" tasks (e.g., edges to image) when it was trained on a very similar task (e.g., segmentation to image) is a huge overclaiming. I proposed the authors take one of the pretrained models and share style transfer results, which is a different task that is not similar to the ones trained on. Unfortunately, the authors did not share these results. Therefore, I am still not convinced, what I do see here is multi-task learning that benefits from text conditioning.
> >
> > 2. The architecture structure assumes pixel-wise correspondences between input-output examples and new test input. This constraint stems from concatenating the images channel-wise in the architecture. The authors claim that using data when there is no pixel correspondences in the tasks will still work. I strongly disagree. Given there is no empirical evidence I will not change my stance here.
> >
> > 3. The architecture structure assumes 3 input images. The authors acknowledge this in the rebuttal but claim changing to ViT can solve this. However, the Stable Diffusion model is not a ViT, and in this case, the authors will not be able to build on it. Therefore I don't see how this helps is currently practical.
> >
> >
> > Given that the authors' have not addressed my concerns, my current recommendation is to reject this paper.

---

> > > ### Author Response · Authors · 2023-08-14
> > >
> > > Thank you for the response, which has helped us to better understand your concerns. We have now followed your request to test Prompt Diffusion not only for style transfer but also under misaligned example input-output pairs. We believe these newly added results have further validated the in-context learning ability of Prompt Diffusion.  Due to the policy, we provide the sharing link with AC and need to wait for further instructions about how it could be shared with reviewers.
> > >
> > >
> > > > I still believe the claim for in-context learning is unwarranted…
> > >
> > > Our experimental results show that the proposed Prompt Diffusion is able to incorporate the visual changes exhibited in the example image pair, aided with text guidance that includes style information or not, to inform how the model will perform image generation given the input query image. The newly added examples per your request, concerning style changes and misalignment, further show that Prompt Diffusion has in-context learning ability in contextualizing its generation based on the visual changes in the example image pairs. In particular, the first row of the newly added examples show that when given a very generic text condition as “A high-quality image.”, Prompt Diffusion can perform style transfer following what is implied in the example image pair.
> > >
> > > We also note that we follow the convention of claiming in-context visual learning in the literature. Painter [2], which follows the setting of [1], claims in-context visual learning on solving high-level understanding and low-level processing tasks, such as semantic segmentation, instance segmentation, depth estimation, keypoint detection, denoising, deraining, and image enhancement. Painter [2] is also trained on a discrete number of tasks. Our work followed their namesake of "in-context learning", but instead focuses on multiple image generation tasks with a diffusion-based backbone. Detailed comparison with Painter [2], in particular how we bring into substantial novelty over its original framework, could be found in our response to Reviewer foSi. We are not overclaiming in the namesake of "in-context learning".
> > >
> > > [1] Bar, Amir, et al. "Visual prompting via image inpainting." Advances in Neural Information Processing Systems 35 (2022): 25005-25017.
> > >
> > > [2] Wang, Xinlong, et al. "Images speak in images: A generalist painter for in-context visual learning." Proceedings of the IEEE/CVF Conference on Computer Vision and Pattern Recognition. 2023.
> > >
> > > > Claiming that in-context learning emerges on "unseen" tasks (e.g., edges to image) when it was trained on a very similar task (e.g., segmentation to image) is a huge overclaiming.
> > >
> > > We disagree we had overclaimed the ability of Prompt Diffusion:
> > >
> > > - Tasks look similar from human perception does not necessarily mean the model trained on one task could easily transfer to the others. We conducted one ablation study in Figure 10 and show that a ControlNet finetuned on an individual task easily fails to generate high-quality images on other tasks. Therefore, it could be misleading to claim two tasks “very similar” by just visual experience.
> > >
> > > - We note the newly added results show without style-specific guidance from texts, Prompt Diffusion successfully performs style transfer (following your recommendation), which is a task distinct from the six training tasks used to train Prompt Diffusion. This result further solidifies our in-context task unseen generalization ability, and we will happily include this new result into our final draft
> > >
> > > > Therefore, I am still not convinced, what I do see here is multi-task learning that benefits from text conditioning.
> > >
> > > We do not think multi-task learning conflicts with in-context learning. As highlighted and validated in the T5[3] paper, multi-task learning serves as a foundational step towards in-context learning in large language models. Training across multiple tasks with expansive data samples is crucial for the emergence of in-context learning. We will clarify this interconnection in the revised paper, but our additional clarifications and new results provided above shall have justified “in-context” learning well.
> > >
> > > [3] Raffel, Colin, et al. "Exploring the limits of transfer learning with a unified text-to-text transformer." The Journal of Machine Learning Research 21.1 (2020): 5485-5551.
> > >
> > > > I proposed the authors take one of the pretrained models and share style transfer results, which is a different task that is not similar to the ones trained on.
> > >
> > > Sorry we did not fully understand your request in the last round - and we have conducted the required experiment now. As previously mentioned, we have kindly requested AC to help pass along the image style transfer results.

---

> > > ### Author Response · Authors · 2023-08-14
> > >
> > > > This constraint stems from concatenating the images channel-wise in the architecture. The authors claim that using data when there is no pixel correspondences in the tasks will still work. I strongly disagree.
> > >
> > > Using pixel-aligned input-output pairs is a standard setting not only in previous in-context visual learning works[1,2], but also in traditional vision tasks (segmentation, keypoint detection, image processing, etc.). Note in our relevant prior arts, all the tasks in Painter [2] also only consider pixel aligned inputs, due to the positional encoding layer inside their transformer architecture. The practical significance of the misalignment case is unclear to us, i.e., why one would desire output images misaligned with their corresponding inputs.
> > >
> > > Nevertheless, as per your request, we have provided image generation results with specifically misaligned input-output pairs. Please check our new results, which (although we feel as artificial) indeed shows our flexible “in-context” generalization to pixel-unaligned input-output tasks.
> > >
> > > > The authors acknowledge this in the rebuttal but claim changing to ViT can solve this. However, the Stable Diffusion model is not a ViT, and in this case, the authors will not be able to build on it.
> > >
> > > We believe there is a misunderstanding on how the ViT backbone is applied to our framework and we would like to clarify the details. Stable Diffusion does not need to be a ViT. Either ControlNet or Prompt-Diffusion constructs separate branches to capture conditions, where the extracted condition feature is aggregated with Stable Diffusion’s feature. Therefore, the design principle of this additional branch does not conflict with Stable Diffusion itself. Under this circumstance, we believe that adopting a ViT based image encoder tailored for the newly constructed branch could potentially enable the ability of flexible token length.
> > >
> > > In fact, we are actually building this now and see no blocker. As it is an ongoing work and is also beyond the scope of this paper, we are refraining ourselves from further discussing it during the review.

---

> > > > ### Comment · Area_Chair_tutv · 2023-08-18
> > > > **Reposting reviewer Gb8x's message**
> > > >
> > > > Dear Authors and Reviewer Gb8x,
> > > >
> > > > I observed that the most recent message from Reviewer Gb8x was intended for the authors but inadvertently omitted including 'Authors' as one of the recipients. Therefore, I am reposting the response directly here, with the expectation that the authors may wish to utilize the remaining time to provide a further response.
> > > >
> > > > >For unseen tasks, I see this as the first evidence of the model extending to "unseen tasks". Despite that, the claim for unseen tasks might still be inaccurate. E.g, even with Style Transfer that is clearly outside of your supervised training data, a quick search within LAION shows me that images with similar structures exist. I would be more comfortable with this "unseen tasks" claim if you downplay it a bit, e.g., whenever you mention unseen tasks continue with "e.g., tasks that are outside of our supervised training dataset". BTW, [1] from above uses a synthetic dataset to demonstrate some generalization capabilities although it can be argued that similar data can exist in LAION 5B as well. I'm open to hearing if you have any thoughts or other suggestions on this as well.
> > > > >
> > > > >
> > > > >As for the non-aligned input-outputs example - my comment was regarding correspondences between the example input and output, and not between the example to new image. For example, consider the task of rotating an input image 90 degrees. The mapping between the two images changes the location of a given pixel. I expect this to fail given the architecture. I propose that you try this and report back. If it fails I expect that you mention this in the limitations section and otherwise provide at least qualitative evidence that it works.
> > > >
> > > > AC

---

> > > > > ### Author Response · Authors · 2023-08-19
> > > > >
> > > > > We thank Reviewer Gb8x for the further response and greatly appreciate AC’s mediating in the rebuttal process. We provide more clarifications below.
> > > > >
> > > > > > 1st comment
> > > > >
> > > > > We commit to be cautious in our next revision when we discuss the generalization ability on “unseen tasks”. We will highlight our style transfer and mis-aligned generalization results as representative unseen tasks that are distinct from training tasks.
> > > > >
> > > > > We note that generating out-of-distribution data is in general a very hard problem. We further elaborate this point in our response below.
> > > > >
> > > > > > 2nd comment
> > > > >
> > > > > We first note that images with unusual angles, such as rotated with 90 degrees, are very rare in the dataset used to train Stable Diffusion, the pretrained backbone that Prompt Diffusion relies on. We also note that image physical modifications, such as rotation and resizing, are not included in our training tasks. Therefore, Stable Diffusion itself would have a very limited ability to generate rotated images that would be considered as out-of-distribution samples. Consequently, Prompt Diffusion is not expected to be able to follow the rotation instruction, regardless of whether that instruction is contextualized in the visual prompt, in the text, or in both.
> > > > >
> > > > > To confirm our analysis, we have conducted three additional experiments: First, we rerun the first Style Transfer example except for rotating the output image in the visual prompt by 90 degrees. Second, we rerun the first Style Transfer example except for adding “rotate the image by 90 degrees” into the text instruction. Third, we add the 90-degree rotation instruction into both the visual and text prompts.
> > > > >
> > > > > As shown in the results, which can be found in the second page of the same Google Doc previously shared with the Area Chair, the output images of Prompt Diffusion successfully follow the style changes implied by the visual prompt, but fail to follow the rotation instructions implied by the visual prompt and/or explicitly given by the text prompt. While we acknowledge not being able to generate rotated images is a limitation of Prompt Diffusion, this limitation is rooted at the Stable Diffusion backbone that lacks the ability to generate rotated images and hence does not weaken our claim of the in-context learning ability of Prompt Diffusion. We will elaborate this point when discussing the limitations in our revision.
> > > > >
> > > > > We would like to acknowledge that, in our observations, models trained specifically for image generation tasks have not exhibited generalization capabilities towards image physical transformation tasks, such as rotations. If you are aware of any pertinent references in this regard, we would greatly appreciate your insights and recommendations.

---

> > > > > ### Author Response · Authors · 2023-08-21
> > > > > **Last Minute Comment**
> > > > >
> > > > > We would greatly appreciate it if you could review our new response. We believe that we have effectively addressed all of your previous concerns. We actively stand by for the last 3 hours of the discussion phase.

---

> > > > > > ### Comment · Reviewer_Gb8x · 2023-08-21
> > > > > > **Response to authors**
> > > > > >
> > > > > > Thank you for agreeing to include these two limitations in the next version of the paper. Given that, I've decided to update my score.

---

> > > > > > > ### Author Response · Authors · 2023-08-21
> > > > > > > **Reply**
> > > > > > >
> > > > > > > We thank Reviewer Gb8x for reconsidering the rating, We appreciate your constructive suggestions and are pleased that we could address your concerns.

---

### Official Review · Reviewer_vt4T · 2023-07-05

**Soundness:** 3 good
**Presentation:** 4 excellent
**Contribution:** 4 excellent
**Rating:** 7
**Confidence:** 4

**Summary:**

The paper proposes a novel prompt design and model called Prompt Diffusion, for in-context learning in vision-language tasks. The vision-language prompt is designed by replacing text examples with paired image examples and the text query with an image query. The paper conducts extensive experiments to demonstrate Prompt Diffusion as a strong versatile vision-language foundation model that is capable of in-context learning.

**Strengths:**

This paper pioneered a new and timely problem setting: extending in-context learning beyond LLMs, to text2image diffusion models. The proposed model, Prompt Diffusion, integrates the learning of multiple tasks into one vision-language foundation model, and it acquires in-context learning ability by learning across multiple tasks and generalizes effectively across new, unseen tasks.

The framework consists of two main components: a vision-language prompt and a diffusion model. The vision-language prompt replaces text examples with paired image examples and the text query with an image query, allowing for a new input-output pair format that could generalize the input-output configuration of most vision-language tasks. The implementation borrowed ideas from ControlNet. The diffusion model then takes the vision-language prompt as input and is trained jointly on six different tasks using these prompts. The resulting Prompt Diffusion model becomes the first diffusion-based foundation model capable of in-context learning.

The model demonstrates high-quality in-context generation for the trained tasks and effectively generalizes to new, unseen vision tasks using their respective prompts. It also demonstrates strong image editing ability. The proposed framework aims to facilitate research into in-context learning for computer vision.


**Weaknesses:**

While this pilot study is highly interesting, it is hard to assess if the diffusion model has the potential to achieve truly generalizable “in-context learning” as LLMs do. Currently, the model is only trained jointly across six pre-defined tasks, and those tasks are similar dense prediction type. The limited diversity of tasks evaluated in this paper constitutes an important hurdle to assess the work’s true value.

It is also hard to predict whether expanding the range of joint tasks would scale up Prompt Diffusion’s in-context learning capability.  Additionally, the model has poor robustness in handling misaligned query images (Figure 7). I am curious how the authors think this can be fixed: shall we expect to encompass more diverse training tasks, training data, etc?

The current generation results of high-fidelity, real-life images look underwhelming. It is noted that the authors used SD v1.5 backbone. I’ll be curious to see if in-context learning can work better if starting from SD v2.1.


**Questions:**

See above

**Limitations:**

The authors have thoroughly discussed limitations in Section 5.

---

> ### Author Rebuttal · Authors · 2023-08-07
>
> We thank Reviewer vt4T for acknowledging the innovativeness of our approach to the problem and appreciating the results of our experiments. Below, we address your questions and concerns:
>
> - We acknowledge the potential benefits of utilizing a larger and more diverse in-context learning dataset to strengthen Prompt-Diffusion. At present, our version is constrained by the available data, which incorporates as few as 6 tasks.
>
> - In Figure 7, we demonstrated that the generation quality is limited when the task is misaligned with our training tasks, and the input text lacks sufficient information. We believe that incorporating more tasks will render the model more adaptable to out-domain tasks.
>
> - As stated in section 5, "As our base dataset is synthesized by Stable Diffusion with proper filtering, it is difficult for the model to generate high fidelity, real-life images." We attribute the underwhelming real-life image generation partly to the base dataset from Instruct Pix2Pix, which is a critical factor since the finetuning dataset is entirely generated by Stable Diffusion and then filtered by CLIP. We firmly believe that a more diverse and high-quality dataset containing real-life images would significantly address this issue. In our preliminary experiments, we did not observe substantial differences between using SD v1.5 and v2.0.

---

### Official Review · Reviewer_7sFz · 2023-07-05

**Soundness:** 3 good
**Presentation:** 4 excellent
**Contribution:** 3 good
**Rating:** 8
**Confidence:** 5

**Summary:**

Prompt Diffusion is a novel framework that enables in-context learning in diffusion-based generative models. It consists of a vision-language prompt and a diffusion backbone, which is trained jointly on six different tasks using their respective prompts. The resulting Prompt Diffusion model becomes the first diffusion-based vision-language model capable of in-context learning, demonstrating high-quality in-context generation for the trained tasks while generalizing to unseen vision tasks.

**Strengths:**

1) The proposed vision-language prompt design is novel, which replaces texts with paired images and replaces the text query with an image query. This design allows for a new input-output pair format that is able to generalize the input-output configuration for most vision-language tasks.

2) The Prompt Diffusion model is the first diffusion-based versatile vision-language foundation model capable of in-context learning. The model acquires in-context learning ability by multi-task learning pre-training, so as to generalize “zero-shot” to additional new tasks with no further adaptation

3) The authors conducted extensive experiments to illustrate their capablity of in-context learning. An ablation study is conducted to investigate the effects of vision-language prompt, which further supports the effectiveness of the proposed design.

4) The model also yields compelling text-guided image editing results, which demonstrate the controllability of the model in generating images based on text guidance.

**Weaknesses:**

1) I would be cautious to call the current model to have " strong in-context learning ability", since the unseen tasks tested in this paper are still similar to the known ones. For example, "HED2Image" task versus "Canny2Image" and "Scribble2Image" tasks. The paper would be strengthened greatly if more diverse unseen tasks could be learned in context.

2) For text encoding, the authors used a pre-trained CLIP. I wonder that if replace CLIP with an LLM encoder module, some of which have its own in-context learning ability in language domain, will enhancethe in-context learning ability?  This is explored by a rencent work: "LLM-grounded Diffusion: Enhancing Prompt Understanding of Text-to-Image Diffusion Models with Large Language Models".

3) The current framework follows instructive pix2pix in most cases and the implementation leverages ControlNet straightforwardly. However, this is not my major concern due to the unconventionality and novelty of the proposed method.

**Questions:**

Please refer to the weaknesses.

**Limitations:**

Yes.

---

> ### Author Rebuttal · Authors · 2023-08-07
>
> We thank Reviewer 7sFz for evaluating the significance of our work and providing encouraging feedback. We provide more clarifications below.
>
> - We commit to exercising greater prudence in claiming the ability of in-context learning in our revised manuscript.
>
> - Appreciate your insight. Indeed, integrating stronger text encoding models, such as LLMs, might potentially enhance the in-context learning ability of our model. Currently, we are actively investigating the application of LLMs within our framework. Thanks for pointing out the recent work. We acknowledge this noteworthy work and intend to cite it while discussing its potential future implementation in our next revision.
>
> - Our data originates from the IP2P dataset, and we acknowledge that employing a larger and more diverse in-context learning dataset could further augment the capabilities of Prompt-Diffusion.

---

> > ### Comment · Reviewer_7sFz · 2023-08-11
> > **About the rebuttal**
> >
> > I am happy with the rebuttal from the authors, which has addressed most of my concerns. Therefore, I have raised the score.

---

> > > ### Author Response · Authors · 2023-08-11
> > >
> > > We greatly appreciate your response and are thankful for your reconsideration in raising the score.

---

### Official Review · Reviewer_foSi · 2023-07-07

**Soundness:** 3 good
**Presentation:** 3 good
**Contribution:** 3 good
**Rating:** 7
**Confidence:** 3

**Summary:**

This work introduces Prompt Diffusion. It is a framework designed to facilitate in-context learning within diffusion-based generative models.
By providing a pair of task-specific example images, such as depth from/to image and scribble from/to image, along with textual guidance,  the designed solution can automatically comprehend the underlying task and replicate the same task on a new query image following the provided text instructions.
Specifically, the authors propose a vision-language prompt that can effectively model various vision-language tasks. The diffusion model, which takes the prompt as input, is trained in a joint manner on six distinct tasks utilizing these prompts.

**Strengths:**

--The paper is well-written and easy to follow.
--The authors have done an excellent job of clearly articulating the motivation behind their work.
--The problem statement is well-defined and the objectives are clearly outlined.


**Weaknesses:**

--  The work appears to be an extension of a previous work titled "A Generalist Painter for In-Context Visual Learning". The design principles and motivations are similar, while there are notable differences that distinguish this work from the previous one.
-- Differences from Prior Work: The authors have introduced text commands into prompts, which adds a new dimension to the model and potentially increases its versatility and applicability. Additionally, the foundation model used in this work is different from the one used in the prior work. This change could potentially lead to different performance characteristics and should be explored in more detail.



**Questions:**

N.A.

**Limitations:**

Overall, this paper presents an interesting extension to previous work in the field. The clarity of the writing and the strong motivation make it a valuable contribution. However, a more detailed comparison with the prior work could strengthen the paper and provide more insights into the novelty and significance of the proposed approach.

---

> ### Author Rebuttal · Authors · 2023-08-07
>
> We thank Reviewer foSi for providing positive feedback. We hereby provide a more detailed comparison between our work and Painter.
>
> - **Task.** Painter follows [1] and focuses on solving high-level discriminative and low-level processing tasks, such as semantic segmentation, instance segmentation, depth estimation, keypoint detection, denoising, deraining, and light enhancement. Our Prompt Diffusion model, instead, further explores condition-guided image generations.
> - **Prompting.** Painter follows [1] and stitches example images, query images, and output images into one large image as the input prompt, which will dramatically increase the computational cost, especially in high-resolution cases. However, we use concatenation in the channel dimension and split the first stage encoding of example images and query images via two independent stacked convolutional layers. Note our model could generate high-resolution, photorealistic images, such as the 512x512 images shown in the paper.
> - **Model Architecture.** Painter follows [1] to formulate the problem as a demasking problem and a Transformer-based image inpainting model (ViT) is trained to predict the masked output tokens. Our work formulated the problem as an image generation problem and the state-of-the-art diffusion model, Stable Diffusion, is our backbone model for finetuning.
> - **Modality.** Painter only allows image-visual prompts, while our work supports multi-modal vision-language prompts. Quoting your comment, the external dimension brings more versatility and applicability to our model.
>
> Drawing upon the aforementioned statements, we point out that Prompt Diffusion explores a different direction of in-context learning in visual models, instead of "an extension of a previous work, Painter." We believe that both works offer distinct and promising viewpoints on in-context visual learning.
>
> References:
>
> [1] Bar, Amir, et al. "Visual prompting via image inpainting." Advances in Neural Information Processing Systems 35 (2022): 25005-25017.

---

> > ### Comment · Reviewer_foSi · 2023-08-15
> > **Response to authors' rebuttal**
> >
> > I am glad that my previous concerns have been addressed. I will change my rating. Thank you.

---

> > > ### Author Response · Authors · 2023-08-17
> > >
> > > We thank Reviewer foSi for taking the time to review our response and reconsider your rating. We greatly appreciate your constructive feedback and are pleased that we could address your concerns.

---

### Comment · Area_Chair_tutv · 2023-08-11
**Reviewer-author discussion**

Dear Reviewers,

Please take a moment to read the authors' responses. Your insights and feedback are crucial in ensuring a comprehensive evaluation. Thanks.

AC

---

### Decision · Program_Chairs · 2023-09-21

**Decision:**

Accept (spotlight)

**Comment:**

The reviewers acknowledge the paper's novel and intriguing problem setting involving visual prompting conditioned on both images and text. The introduction of pretrained diffusion models to incorporate visual and text cues is highlighted as a strong point, enabling improved visual quality and text-guided diffusion. The novel vision-language prompt design, which replaces text with image pairs and queries, is emphasized for its potential to generalize across various vision-language tasks. The reviewer appreciates the model's unique in-context learning ability achieved through multi-task learning pre-training, allowing it to adapt to new tasks seamlessly. The extensive experiments and ablation study conducted to validate the proposed approach's effectiveness further contribute to the paper's strength.